# The Effect of Low HBV-DNA Viral Load on Recurrence in Hepatocellular Carcinoma Patients Who Underwent Primary Locoregional Treatment and the Development of a Nomogram Prediction Model

**DOI:** 10.3390/microorganisms12050976

**Published:** 2024-05-13

**Authors:** Yiqi Xiong, Ziling Wang, Jiajun Liu, Kang Li, Yonghong Zhang

**Affiliations:** 1Interventional Therapy Center for Oncology, Beijing You’an Hospital, Capital Medical University, Beijing 100069, China; x790152766@163.com (Y.X.);; 2Research Center for Biomedical Resources, Beijing You’an Hospital, Capital Medical University, Beijing 100069, China

**Keywords:** hepatocellular carcinoma, HBV-DNA, TACE, ablation, nomogram, recurrence

## Abstract

(1) Background: HBV-DNA is an essential clinical indicator of primary hepatocellular carcinoma (HCC) prognosis. Our study aimed to investigate the prognostic implication of a low load of HBV-DNA in HCC patients who underwent local treatment. Additionally, we developed and validated a nomogram to predict the recurrence of patients with low (20–100 IU/mL) viral loads (L-VL). (2) Methods: A total of 475 HBV-HCC patients were enrolled, including 403 L-VL patients and 72 patients with very low (<20 IU/mL) viral loads (VL-VL). L-VL HCC patients were randomly divided into a training set (N = 282) and a validation set (N = 121) at a ratio of 7:3. Utilizing the Lasso–Cox regression analysis, we identified independent risk factors for constructing a nomogram. (3) Results: L-VL patients had significantly shorter RFS than VL-VL patients (38.2 m vs. 23.4 m, *p* = 0.024). The content of the nomogram included gender, BCLC stage, Glob, and MLR. The C-index (0.682 vs. 0.609); 1-, 3-, and 5-year AUCs (0.729, 0.784, and 0.783, vs. 0.631, 0.634, the 0.665); calibration curves; and decision curve analysis (DCA) curves of the training and validation cohorts proved the excellent predictive performance of the nomogram. There was a statistically significant difference in RFS between the low-, immediate-, and high-risk groups both in the training and validation cohorts (*p* < 0.001); (4) Conclusions: Patients with L-VL had a worse prognosis. The nomogram developed and validated in this study has the advantage of predicting patients with L-VL.

## 1. Introduction

Primary liver cancer is one of the most common malignant tumors and a leading cause of cancer-related deaths in China. The predominant pathological type of liver cancer is hepatocellular carcinoma (HCC), which accounts for 75~85% [1]. HCC ranks fifth among malignant tumors in China, with the second-highest mortality rate [2]. Up to now, there have been many treatment modalities for HCC, such as hepatectomy, liver transplantation, ablation therapy, interventional therapy, radiotherapy, biological therapy, and molecular targeted therapy [3]. Ablation treatment stands out as a leading option for early hepatocellular carcinoma [4]. Its efficacy is nearly on par with surgery and liver transplantation, boasting reduced expenses and quicker postoperative recovery [5]. 

Nevertheless, regrettably, the recurrence rate following ablation remains notable, with a five-year recurrence rate ranging from 50% to 70% [6,7]. For patients diagnosed with intermediate HCC, the Barcelona Clinical Liver Cancer (BCLC) guidelines recommend transarterial chemoembolization (TACE) as the gold standard treatment method, but the median progression-free survival (mPFS) after treatment is only eight months [8,9]. Due to the poor prognosis, HCC has become a significant challenge in global healthcare [10].

In China, more than 80% of liver cancer patients are co-infected with hepatitis B virus (HBV) [11]. The HBV viral load is considered an essential factor in predicting tumor recurrence [12,13,14], and in most HBV-related HCC patients, a high viral load is usually associated with poor prognosis [15,16]. Preoperative HBV-DNA exceeding the quantitative lower limit (≥200 IU/mL) is an independent risk factor for HBV reactivation. Prior research has demonstrated that despite low HBV-DNA levels prior to hepatectomy in HBV-related HCC patients, their propensity for HCC recurrence remains substantial [17], which indicates that timely consideration of antiviral therapy is necessary [18,19]. Antiviral therapy can improve the prognosis of the disease [20]. The quantitative measurement of HBV-DNA can reliably reflect the replication status of the HBV virus in individuals with chronic hepatitis B, offering significant utility in monitoring treatment efficacy, devising therapeutic regimens, and determining treatment endpoints [21].

At present, the routine method for HBV-DNA detection in China has a lower detection limit set at 200 IU/mL. While our center’s detection capability extends to a minimum value of 100 IU/mL, it remains challenging to meet the clinical diagnostic requirements fully. With the advent of high-sensitivity real-time quantitative PCR assays, the detection limit has been reduced to 10~20 IU/mL, which has the advantages of high sensitivity, robust specificity, and low carryover contamination. However, it is costly and has a complex detection procedure [22,23,24,25]. Therefore, it is worth considering whether it is necessary to use high-sensitivity detection methods before intervention to screen out patients with low viral loads.

Several prognostic markers associated with HBV, including the Child–Pugh classification, ALT levels, PVTT presence, tumor number, tumor diameter, treatment type, and lipid levels, have been identified to correlate with prognosis [26]. However, the relevance of recurrence and a low DNA load remains to be determined, as well as a scarcity of biomarkers or nomograms for low viral loads. Consequently, the purpose of our study is to differentiate the prognosis of patients with low (20–100 IU/mL) viral loads (L-VL) and very low (<20 IU/mL) viral loads (VL-VL) after undergoing TACE combined with ablation. Additionally, we developed and validated a nomogram to identify high-risk patients with poor prognosis, which enhances clinicians’ comprehension of the impact of preoperative viral load levels on the survival outcomes of HBV-HCC patients and aids physicians in making decisions.

## 2. Materials and Methods

### 2.1. Patients

This retrospective cohort study reviewed 475 HBV-HCC patients who underwent TACE combined with ablation at Beijing Youan Hospital from January 2014 to December 2021, including 72 VL-VL patients and 403 L-VL patients. The diagnostic criteria for HCC followed the guidelines set forth by the American Association for the Study of Liver Diseases (AASLD) [27]. To establish a more reliable and robust model, 403 L-VL patients were randomly allocated in a 7:3 ratio into a training set (N = 282) and a validation set (N = 121). The inclusion criteria for patients were as follows: (1) patients with preoperative HBV-DNA level 20–100 IU/mL or <20 IU/mL; (2) early liver cancer patients who achieved complete remission after TACE combined with ablation; (3) patients with Child–Pugh A or B liver function; (4) all patients who had not received any other treatment before ablation.

Exclusion criteria included (1) a diagnosis of other malignant diseases within the past five years; (2) autoimmune liver disease; (3) previous treatment with other drugs, such as Chinese patent medicine, immunomodulatory drugs, glucocorticoid therapy, or other immunosuppressive treatments; (4) incomplete clinical follow-up data.

Our study was approved by the Ethics Committee of Beijing Youan Hospital, Capital Medical University, and conducted following the standards of the Helsinki Declaration. The Ethics Committee considered the study low-risk and waived the requirement for informed consent from patients.

### 2.2. Clinicopathologic Characteristics

Clinical pathological characteristics of preoperative patients were collected, which was composed of age, gender, hypertension, diabetes, cirrhosis, smoking, drinking, HBV-DNA viral load, cirrhosis, the Child–Pugh score, BCLC, number of tumors, tumor size, white blood cells (WBCs), eosinophils, basophils, red blood cells (RBCs), neutrophil-to-lymphocyte ratio (NLR), monocyte-to-lymphocyte ratio (MLR), platelet-to-lymphocyte ratio (PLR), hemoglobin (Hb), alanine aminotransferase (ALT), aspartate aminotransferase (AST), total bilirubin (TBIL), direct bilirubin (DBIL), albumin (Alb), globulin (Glob), globulin-to-lymphocyte ratio (GLR), alkaline phosphatase (ALP), prealbumin (Palb), blood urea nitrogen (BUN), prothrombin time (PT), prothrombin activity (PTA), prothrombin ratio (PTR), international normalized ratio (INR), activated partial thromboplastin time (APTT), fibrinogens (Fib), thrombin time (TT), and alpha-fetoprotein (AFP).

### 2.3. Treatment Received 

#### 2.3.1. TACE Procedure

Two interventional radiologists, each boasting five years of expertise in this technique, conducted the transarterial chemoembolization (TACE) procedure. Using local anesthesia, they accessed the right femoral artery. A catheter was subsequently advanced into the hepatic artery via percutaneous puncture and linked to a high-pressure injector under digital subtraction angiography (DSA) guidance to visualize intrahepatic arteries, left and right hepatic arteries, and their branches. Employing a selective/super-selective technique, a highly flexible coaxial microcatheter was navigated to the tumor-feeding artery to infuse a mixture of doxorubicin and iodized oil. Following connection to a high-pressure injector for imaging, embolization was executed using embolic materials, exemplified by gelatin sponge or polyvinyl alcohol particles, until the complete cessation of blood flow in the vessel. Based on the patient’s white blood cell count, platelet count, and liver function, we determined the drug dosage. The completion of the procedure was indicated by intratumoral vessel occlusion, complete embolic agent deposition, and the absence of tumor enhancement.

#### 2.3.2. Ablation Procedure

For the ablation procedure, local ablation was conducted within two weeks post-TACE, guided by three-phase computed tomography (CT) and magnetic resonance imaging (MRI). The application of electrodes depended on the size of the tumor. We supplemented standard disinfection and local anesthesia at the puncture site with intravenous analgesia and monitoring anesthesia care. During radiofrequency ablation (RFA), after establishing baseline impedance, power output gradually escalated from 80 watts to 200 watts until maximal impedance was achieved. Cold saline infusion into the electrode cavity maintained a tip temperature below 20 °C. Expanding the ablation zone by 0.5–1.0 cm ensured thorough ablation for achieving complete tumor eradication; incomplete ablation was flagged otherwise. Heating the electrode to 90 °C–100 °C followed by withdrawal mitigated postoperative bleeding and tumor implantation along the needle tract. Immediate post-ablation enhanced CT scans were conducted to assess technical success and identify potential complications.

### 2.4. Follow-Up

After approximately 4–6 weeks of treatment, clinicians assessed responses using CT and MRI scans. According to the RECIST efficacy evaluation criteria, complete response (CR) was confirmed by imaging examination (such as CT, MRI, or ultrasound) as the complete disappearance of all tumor lesions after ablation, with no residual malignant tissue or tumor cells. Patients were examined every three months during the first year after ablation and then every six months after that. Follow-up included blood tests, liver function tests, and imaging examinations to detect tumor recurrence. The primary endpoint of this study was recurrence-free survival (RFS), defined as the time from the initiation of autonomous therapy to the first recurrence of liver cancer or death from any cause.

### 2.5. Statistical Analysis

Continuous variables were presented as mean (standard deviation [SD]) or median (IQR), while categorical variables were presented as frequency distribution (n, %). Independent-sample *t*-test, Mann–Whitney–Wilcoxon, and Pearson chi-square tests were employed for group comparisons. RFS was estimated by Kaplan–Meier and compared with the log-rank test. The nomogram was constructed using the predictive factors selected by multivariate Cox regression followed by Lasso regression. The performance of the nomogram was assessed using the C-index, area under the curve (AUC) of the receiver operating characteristic (ROC) curves, calibration curves, and decision curve analysis (DCA) curves. Based on the scores derived from curve plots, the patient population was stratified into low-risk, intermediate-risk, and high-risk groups, with KM curve analysis to observe prognosis differences.

Statistical analysis was performed using SPSS (version 26.0, IBM, Armonk, NY, USA) and R software (version 4.1.3). All statistical tests were conducted at a significance level of 0.05. Additionally, 95% confidence intervals and *p*-values were displayed when calculating intergroup mean differences.

## 3. Results

### 3.1. Patient Characteristics

From January 2014 to December 2021, a total of 475 patients who underwent TACE combined with ablation in Beijing Youan Hospital were screened in our study, containing 403 L-VL patients and 72 VL-VL patients. The baseline characteristics of the two groups of patients were similar, with nonsignificant discrepancies in variables such as gender (mostly male), the Child–Pugh score (mostly A), the number of tumors (primarily solitary), tumor size (<3 cm), alanine aminotransferase (ALT), aspartate aminotransferase (AST), total bilirubin (TBIL), direct bilirubin (DBIL), and several other important variables (Table 1).

Patients with an HBV-DNA viral load of 20–100 IU/mL were randomly divided into a training set (N = 282) and a validation set (N = 121) at a ratio of 7:3 to establish a more reliable and robust model, and there was no statistical significance in baseline characteristics between the two groups (Table 2). In both the training and validation sets, the majority of patients were male (79.8% vs. 75.2%, *p* = 0.372), with an average age ≤75 years (57.9 ± 8.48 vs. 57.4 ± 8.96, *p* = 0.609). Most patients were Child–Pugh A (73.8% vs. 72.7%, *p* = 0.927), which signifies healthy liver function. Regarding tumor characteristics, most tumors were solitary (72.7% vs. 64.5%, *p* = 0.124), and tumor size was less than 3 cm (67.7% vs. 70.2%, *p* = 0.703).

### 3.2. Efficacy

On July 30, 2023, the median follow-up time was 3.47 years. The KM curve displayed that patients with L-VL had significantly shorter RFS than those with VL-VL (38.2 m vs. 23.4 m, *p* = 0.024). The 1-, 3-, and 5-year RFS rates in the L-VL group were 70.4%, 38.0%, and 26.7%, whereas those in the VL-VL group were 80.6%, 51.3%, and 41.4%, respectively. As depicted in Figure 1, patients with L-VL exhibited worse prognoses and higher recurrence rates, highlighting the necessity of establishing a predictive model to identify high-risk populations and enable timely intervention.

### 3.3. Independent Prognostic Factors Based on Lasso–Cox Regression

The training set was used to build the model. Lasso regression was utilized to screen parameters, and the change characteristics of these variable coefficients are shown in Figure 2A. Applying the 10-fold cross-validation method to iterative analysis yielded a model with excellent performance and the fewest variables when λ was 0.059 (Figure 2B). Seven variables, namely age, gender, BCLC, ablation treatment, GLR, Glob, and MLR, were selected through Lasso regression. Based on the parameters selected by Lasso regression, a multivariate Cox regression model was further established (Table 3). The result manifested that gender (HR: 0.45, 95% CI: 0.30–0.69), BCLC (HR: 1.6, 95% CI: 1.30–2.05), Glob (HR: 1.03, 95% CI: 1.00–1.06), and MLR (HR: 2.27, 95% CI: 1.14–4.52) were independent predictors of RFS (Table 3).

### 3.4. Development of Nomogram

Through Lasso–Cox regression analysis, independent predictive factors were identified to construct a nomogram (Figure 3). The C-index in the training set was 0.682 (95% CI: 0.644–0.719). The ROC curves of the training cohort displayed AUCs of 0.729, 0.784, and 0.783 for 1-, 3-, and 5-year RFS, respectively (Figure 4A). Calibration plots of 1-, 3-, and 5-year RFS reflected good consistency between predicted and observed outcomes (Figure 5A). Additionally, decision curve analysis (DCA) curves were generated to assess the clinical value of the nomogram, showing favorable clinical utility (Figure 6A–C). Based on the nomogram, patients were stratified into low-risk, intermediate-risk, and high-risk groups. In the training cohort, the low-risk group (*n* = 82) had a median RFS of 38.1 months (95% CI: 25.1–59.5) with 1-, 3-, and 5-year RFS rates of 79.1%, 50.7%, and 33.8%, respectively. The intermediate-risk group (*n* = 109) had a median RFS of 20.7 months (95% CI: 16.6–26.2) with 1-, 3-, and 5-year RFS rates of 66.0%, 33.2%, and 22.3%, respectively. The high-risk group (*n* = 91) had a median RFS of only 13.6 months (95% CI: 11.1–20.9), with 1-, 3-, and 5-year RFS rates of 55.7%, 23.9%, and 16.7%, respectively. The three groups had no statistically significant distinction (*p* < 0.0001) (Figure 7A).

### 3.5. Validation of the Nomogram

To enhance the robustness of the nomogram, we undertook an internal validation to confirm its reliability. The validation results were also promising, with a C-index of 0.609 (95% CI: 0.542–0.675). Time-dependent ROC curves of the validation cohort displayed AUCs of 0.631, 0.643, and 0.665 for 1-year, 3-year, and 5-year RFS, respectively (Figure 4B). These results collectively demonstrated the predictive efficacy of the nomogram. The calibration plots of 1-year, 3-year, and 5-year RFS probabilities also reflected good consistency between the predicted and observed outcomes (Figure 5B). Decision curve analysis (DCA) curves exhibited favorable clinical utility (Figure 6D–F). The validation cohort revealed notable discrepancies in RFS across the low-risk (*n* = 34), intermediate-risk (*n* = 44), and high-risk (*n* = 43) groups. Specifically, the high-risk cohort exhibited substantially greater recurrence susceptibility compared to the low- and intermediate-risk cohorts (*p* < 0.0001).

## 4. Discussion

Hepatocellular carcinoma (HCC), one of the most common malignant tumors worldwide, is attributed to hepatitis B virus (HBV) infection in over 80% of HCC patients in our country [11]. HBV viral load is essential in predicting tumor recurrence [12,13,14], with a high viral load often associated with poor prognosis [15,16]. Our research suggested that patients with VL-VL before local treatment had better prognosis than patients with L-VL. Additionally, we established a clinical prediction nomogram for patients with L-VL to help clinicians identify high-risk patients and implement early interventions.

This study included 475 patients, with a median follow-up time of approximately 3.47 years. The mRFS of the VL-VL group was 38.2 months, compared with 23.4 months among the L-VL group (*p* = 0.024). Due to the poor prognosis, a nomogram was established by Lasso–Cox regression and validated through the training cohort (N = 282) and validation cohort (N = 121). Compared to univariate analysis, Lasso regression had advantages in addressing multicollinearity among variables. Based on the Lasso–Cox regression model, we established a nomogram that provides clinicians with an intuitive analysis of individual recurrence risk, guiding the screening of high-risk patients for recurrence after local and regional treatment. Doctors can devise personalized follow-up strategies or treatment plans based on predicted recurrence risk to mitigate recurrence rates and extend overall survival. The scores on the nomogram were obtained by drawing a vertical line at the corresponding total score position, intersecting with the predicted recurrence risk lines to forecast 1-, 3-, and 5-year recurrence-free survival (RFS). The nomogram had good predictive ability, with C-indexes of 0.682 and 0.609 for the training and validation sets, respectively. The high AUC of the time-dependent ROC curves for the training and validation sets confirmed model discrimination, especially at the 3-year and 5-year time points. Calibration curves showed good consistency, and DCA curves confirmed a higher net clinical benefit rate. According to the nomogram, patients were divided into three different risk groups. Significant differences were observed in RFS among the groups (*p* < 0.0001), which emphasized the superior discriminatory ability of our nomogram in identifying patients at risk for postoperative recurrence.

The Lasso–Cox regression found that gender, BCLC stage, Glob, and MLR were associated with HCC recurrence. Male patients displayed an increased risk of recurrence compared to female patients, potentially attributed to gender-specific variations in biological, environmental, and behavioral factors. Notably, androgen receptor activation bolsters the advancement and invasiveness of HCC. In contrast, endogenous estrogen metabolites exert inhibitory effects on tumor proliferation through their antiproliferative, pro-apoptotic, and antiangiogenic properties. The BCLC stage is a clinical staging system developed to assess and classify the severity of liver cancer patients’ conditions and treatment choices. In addition to the tumor characteristics such as size, location, number, and extent of spread, it is also vital to consider the physiological status and hepatic function in HCC patients to formulate personalized treatment plans. Patients with advanced-stage hepatocellular carcinoma (BCLC C and D) generally exhibit a worse prognosis [28].

Globulins, including gamma globulins, antibodies, and glycoproteins, constitute one of the most essential protein groups in the bloodstream. GLB acts as a regulator in the circulatory system by assisting in blood clotting, transporting proteins through lipoproteins, and indicating antibody levels. Elevated levels of globulins can be attributed to chronic inflammatory diseases such as chronic viral or bacterial infections, liver diseases, autoimmune conditions, ulcerative colitis, and kidney diseases. Chronic inflammation is a common cause of various tumors. HCC is an inflammation-related cancer, and sustained chronic inflammatory status is associated with poor prognosis in HCC patients [29]. In some cases, the immune system of liver cancer patients can be compromised, resulting in weakened immune function and increased risk of cancer recurrence. The monocyte-to-lymphocyte ratio (MLR) represents the inflammatory immune microenvironment. Some studies deem that MLR is an important prognostic indicator for liver cancer, possibly due to its relationship with tumor invasiveness and poor liver function. Monocytes aggregate at sites of inflammation and differentiate into M1 and M2 macrophages during inflammation. M2 macrophages stimulate angiogenesis and invasion by secreting the vascular endothelial growth factor (VEGF), which promotes the recurrence and progression of tumors. M3 macrophages secrete chitinase 1-like protein 2 (CHI1L2), which contributes to tumor metastasis through the specific binding to the interleukin (IL)-13 receptor α2 chain (IL-13Rα2) in gastric and breast cancer cell. Additionally, macrophages inhibit the antitumor effects of CD4+ T cells and suppress T-cell proliferation by secreting various molecules such as TGF-β and Arg-1 [30,31,32].

As for viral factors, there is increasing evidence that the HBV viral load is a predictor of HCC recurrence after surgery and negatively impacts overall survival, which affects the prognosis of HBV-related HCC patients in multiple ways. Firstly, the active viral replication of HBV initiates the carcinogenic process directly by increasing the likelihood of HBV-DNA insertion into oncogenes, tumor suppressor genes, and near-cell DNA regulatory elements. Some instability and chronic hepatitis could lead to liver fibrosis and HCC development by triggering an immune response [33]. Like other viruses, HBV induces endoplasmic reticulum (ER) stress. To alleviate ER stress, the unfolded protein response (UPR), including glucose-regulated protein 78 (GRP78), is upregulated in high HBV viral loads [18]. The GRP78 pathway is one of the most essential response factors to disease-related stress and plays a critical role in the stepwise progression of HBV-related HCC [34].

Recent meta-analyses have illustrated that antiviral therapy can reduce the recurrence and mortality of HCC after curative treatment. The primary treatment mechanism is that it prevents HBV replication and reduces HBV reactivation, regardless of HBV viral load [35,36]. Despite the effective reduction in HBV replication and viral load through antiviral therapy, studies have revealed instances that liver cirrhosis can still progress to HCC [37]. Liver cirrhosis patients are more likely to develop liver cancer even with HBV-DNA levels below 100 IU/mL, and lower HBV-DNA levels (<20 IU/mL) mitigate risk. Yet, it has been reported that a high HBV viral load before TACE is linked to frequent exacerbation of hepatitis, and most acute exacerbations respond well to antiviral therapy. International guidelines recommend starting prophylactic antiviral therapy before TACE to minimize the risk of HBV reactivation in HBV-related HCC patients [38].

TACE is a soothing treatment recommended by the BCLC guidelines for intermediate-stage HCC patients. For early-stage HCC patients, TACE can achieve tumor downstaging, shorten ablation time, and improve ablation success rates [39]. Ablation is performed by inserting electrodes into the tumor under ultrasound, CT, or MRI guidance and then generating frictional heat on the tissue by applying high-frequency electrodes to destroy HCC cells [40,41,42]. Both domestic and foreign studies have found that combined treatment with TACE and ablation can improve overall survival and progression-free survival compared to TACE alone [43,44]. Thus, we adopted TACE combined with ablation therapy to improve the quality of life and prolong the overall survival of more HCC patients.

The clinical indicators applied in this study involve demographic statistics, liver function, and inflammatory indicators, enabling a more comprehensive assessment of patients. Notably, the composition of our nomogram is simple and easy to obtain, allowing doctors to assess patients’ conditions promptly and effectively.

Despite the good predictive performance of the nomogram, our study still has some limitations. Firstly, this was a retrospective study that inevitably had some selection bias, and further prospective studies are needed to validate the nomogram. However, our internal validation results reflect the accuracy and reliability of our nomogram. Secondly, our data came from the same hospital. Therefore, large-sample multicenter studies are warranted to verify the funding. Moreover, more research is required to explore our nomogram’s application in patients receiving other treatments. We used a follow-up period of up to seven years to create an accurate and reliable nomogram, which better guides the clinical practice of HCC patients with L-VL.

## 5. Conclusions

Compared to the VL-VL group, the L-VL group had an inferior prognosis. Based on the Lasso–Cox regression analysis, we established a nomogram to predict the recurrence of early-stage HCC patients with low viral loads. The nomogram has a specific guiding significance for screening high-risk groups for post-treatment recurrence, and doctors can develop individualized follow-up strategies or treatment plans for patients based on the predicted recurrence risk to improve long-term prognosis.

## Figures and Tables

**Figure 1 microorganisms-12-00976-f001:**
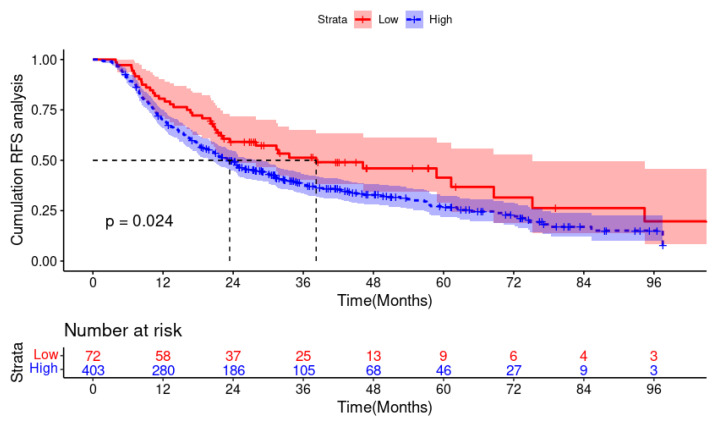
Kaplan–Meier plots for RFS in the low viral load (High) group and very low viral load (Low) group. Abbreviation: RFS: recurrence-free survival.

**Figure 2 microorganisms-12-00976-f002:**
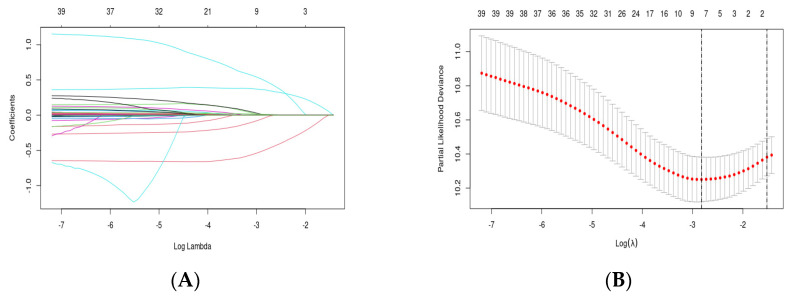
Variable selection based on LASSO (least absolute shrinkage and selection operator) regression: (**A**) characteristics of variable coefficient changes; (**B**) process of selecting the optimal value of parameter λ in the LASSO regression model through cross-validation methods.

**Figure 3 microorganisms-12-00976-f003:**
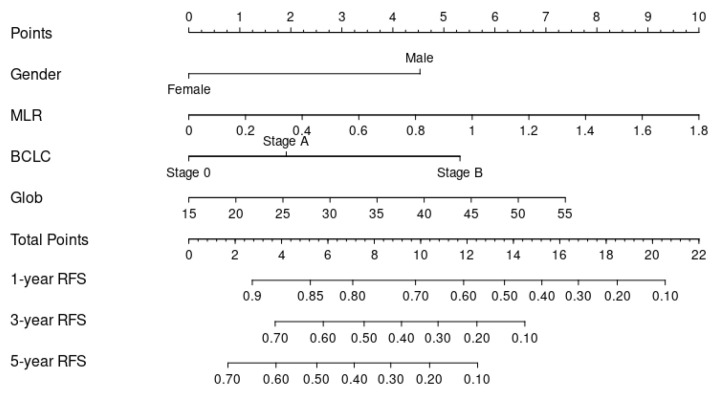
Nomogram, including gender, MLR; BCLC; and Glob for one-, three-, and five-year recurrence-free survival (RFS) in HCC patients with dynamic changes in AFP. The nomogram is valued to obtain the probability of one-, three-, and five-year recurrence by adding up the points identified on the point scale for each variable. Abbreviation: BCLC: Barcelona Clinic Liver Cancer; Glob: globulin; MLR: monocyte-to-lymphocyte ratio.

**Figure 4 microorganisms-12-00976-f004:**
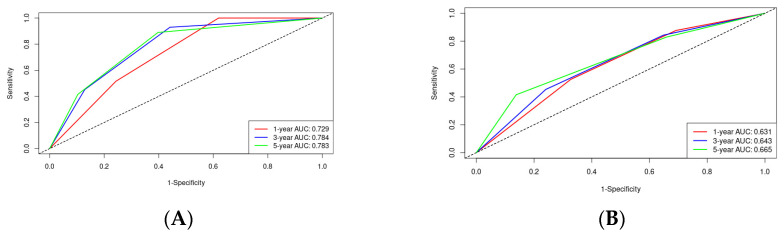
ROC curves of the nomogram in the training cohort and validation cohort: (**A**) the AUCs for 1-, 3-, and 5-year RFS were 0.729, 0.784, and 0.783 in the training cohort; (**B**) the AUCs for 1-, 3-, and 5-year RFS were 0.631, 0.643, and 0.665 in the validation cohort.

**Figure 5 microorganisms-12-00976-f005:**
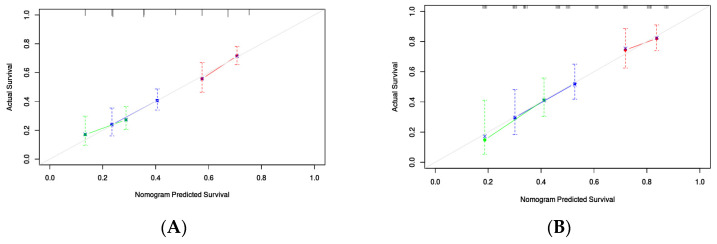
Calibration curves of the nomogram in the training cohort and validation cohort: (**A**) one-, three-, and five-year RFS in the training cohort; (**B**) one-, three-, and five-year RFS in the validation cohort.

**Figure 6 microorganisms-12-00976-f006:**
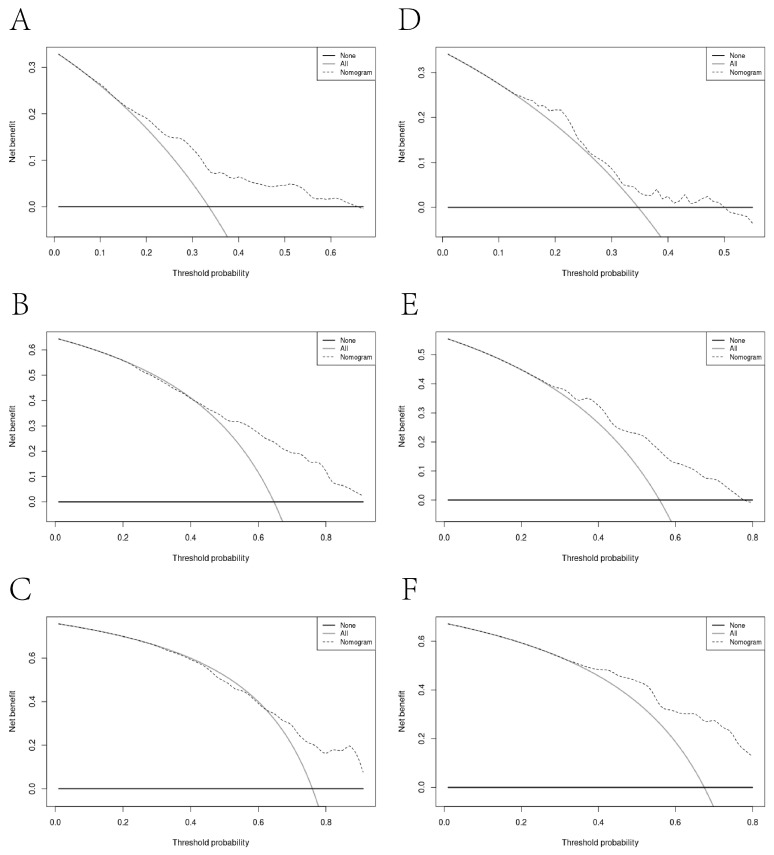
Decision curve analyses for recurrence in the training cohort and validation cohort. The *x*-axis indicates threshold probability, and the *y*-axis indicates the net benefit. Dashed lines: the net benefit of nomogram across a range of threshold probabilities. The horizontal line: no patient relapse. The solid black line: all patients die or relapsed: (**A**) decision curve analysis for one-year RFS in the training cohort; (**B**) decision curve analysis for three-year RFS in the training cohort; (**C**) decision curve analysis for five-year RFS in the training cohort; (**D**) decision curve analysis for one-year RFS in the validation cohort; (**E**) decision curve analysis for three-year RFS in the validation cohort; (**F**) decision curve analysis for five-year RFS in the validation cohort.

**Figure 7 microorganisms-12-00976-f007:**
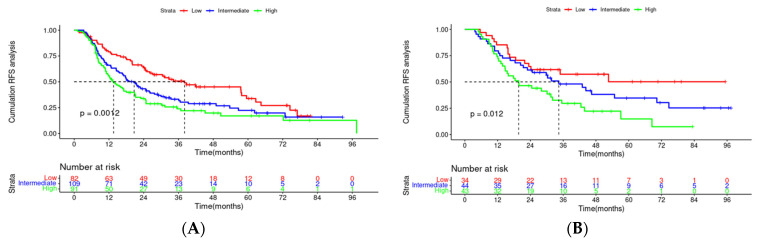
Kaplan–Meier plots of RFS for the low-risk group, medium-risk group, and high-risk group: (**A**) training cohort; (**B**) validation cohort. Abbreviation: RFS, recurrence-free survival.

**Table 1 microorganisms-12-00976-t001:** Demographics and clinical characteristics.

	<20 (N = 72)	20–100 (N = 403)	*p*-Value
Age	55.47 ± 8.70	57.71 ± 8.51	0.047
Gender			0.603
Male	59 (81.9%)	316(78.4%)	
Female	13 (18.1%)	87 (21.6%)	
Hypertension			0.480
Yes	15 (20.8%)	103 (25.6%)	
No	57(79.2%)	300 (74.4%)	
Diabetes			1.000
Yes	16 (22.2%)	91 (22.6%)	
No	56 (77.8%)	312 (77.4%)	
Cirrhosis			1.000
Yes	63 (87.5%)	351 (87.1%)	
No	9 (12.5%)	52 (12.9%)	
Smoking			0.468
Yes	33 (45.8%)	163 (40.4%)	
No	39 (54.2%)	240 (59.6%)	
Drinking			0.527
Yes	26 (36.1%)	127 (31.5%)	
No	46 (63.9%)	276 (68.5%)	
Child–Pugh			0.531
A	56 (77.8%)	296 (73.4%)	
B	16 (22.2%)	107 (26.6%)	
BCLC			0.319
0	31 (43.1%)	137 (34.0%)	
A	31 (43.1%)	207 (51.4%)	
B	10 (13.9%)	59 (14.6%)	
T.N			0.840
Single	52(72.2%)	283 (70.2%)	
Multiple	20 (27.8%)	120 (29.8%)	
T.S			0.622
<30 mm	52(72.2%)	276(68.5%)	
≥30 mm	20 (27.8%)	127 (31.5%)	
WBC	4.82 ± 1.78	5.13 ± 2.31	0.270
NLR	3.03 ± 2.53	3.53 ± 3.26	0.223
MLR	0.35 ± 0.18	0.39 ± 0.23	0.207
RBC	4.13 ± 0.76	4.15 ± 0.62	0.835
Hb	127.78 ± 23.31	129.81 ± 19.48	0.430
PLR	114.05 ± 69.42	111.32 ± 57.89	0.721
ALT	28.07 ± 13.46	27.95 ± 16.18	0.955
AST	30.19 ± 14.39	29.71 ± 11.9	0.761
TBIL	19.45 ± 9.28	20.26 ± 10.55	0.541
DBIL	6.89 ± 3.67	7.18 ± 4.99	0.634
Alb	37.75 ± 4.79	37.04 ± 4.76	0.250
Glob	29.08 ± 5.35	28.44 ± 5.62	0.372
GLR	58.45 ± 53.49	70.03 ± 105.18	0.363
ALP	84.47 ± 26.9	88.44 ± 36.8	0.383
Palb	145.41 ± 64.09	138.08 ± 57.67	0.330
PT	12.82 ± 1.55	12.68 ± 1.63	0.507
PTA	83.3 ± 13.11	85.44 ± 15.99	0.286
INR	1.13 ± 0.13	1.13 ± 0.15	0.664
APTT	33.67 ± 3.99	33.52 ± 4.86	0.801
Fib	2.61 ± 0.76	2.8 ± 0.94	0.109
TT	15.99 ± 2.12	15.94 ± 2.20	0.848
AFP	180.57 ± 609.06	315.8 ± 1882.44	0.546

Abbreviation: BCLC: Barcelona Clinic Liver Cancer; T.N: tumor number; T.S: tumor size; WBC: leukocyte; NLR: neutrophil-to-lymphocyte ratio; MLR: monocyte-to-lymphocyte ratio; RBC: erythrocyte; Hb: hemoglobin; PLR: platelet-to-lymphocyte ratio; ALT: alanine aminotransferase; AST: aspartate aminotransferase; TBIL: total bilirubin; DBIL: direct bilirubin; Alb: albumin; Glob: globulin; GLR: globulin-to-lymphocyte ratio; ALP: alkaline phosphatase; Palb: prealbumin; PT: prothrombin time; PTA: prothrombin activity; INR: international normalized ratio; APTT: activated partial thromboplastin time; Fib: fibrinogens; TT: thrombin time; AFP: alpha-fetoprotein.

**Table 2 microorganisms-12-00976-t002:** Demographics and clinical characteristics for training and validation sets.

	Training (N = 282)	Validation (N = 121)	*p*-Value
Age	57.9 ± 8.48	57.4 ± 8.95	0.609
Gender			0.372
Male	225 (79.8%)	91 (75.2%)	
Female	57 (20.2%)	30 (24.8%)	
Hypertension			1.000
Yes	72 (25.5%)	31 (25.6%)	
No	210 (74.5%)	90 (74.4%)	
Diabetes			0.636
Yes	66 (23.4%)	25 (20.7%)	
No	216 (76.6%)	96 (79.3%)	
Cirrhosis			0.971
Yes	245 (86.9%)	106 (87.6%)	
No	37 (13.1%)	15 (12.4%)	
Smoking			0.154
Yes	121 (42.9%)	42 (34.7%)	
No	161 (57.1%)	79 (65.3%)	
Drinking			0.279
Yes	94 (33.3%)	33 (27.3%)	
No	188 (66.7%)	88 (72.7%)	
Child–Pugh			0.927
A	208 (73.8%)	88 (72.7%)	
B	74 (26.2%)	33 (27.3%)	
BCLC			0.692
0	98 (34.8%)	39 (32.2%)	
A	141 (50%)	66 (54.5%)	
B	43 (15.2%)	15 (13.2%)	
T.N			0.124
Single	205 (72.7%)	78 (64.5%)	
Multiple	77 (27.3%)	43 (35.5%)	
T.S			0.703
<30 mm	191 (67.7%)	85 (70.2%)	
≥30 mm	91 (32.3%)	36 (29.8%)	
WBC	5.06 ± 2.19	5.31 ± 2.58	0.351
NLR	3.32 ± 3.00	4.00 ± 3.77	0.083
MLR	0.38 ± 0.22	0.41 ± 0.26	0.298
RBC	4.14 ± 0.63	4.17 ± 0.59	0.621
Hb	130 ± 20.0	130 ± 18.3	0.935
PLR	109 ± 53.7	117 ± 66.6	0.238
ALT	28.1 ± 17.4	27.6 ± 12.9	0.73
AST	30.2 ± 12.8	28.5 ± 9.33	0.134
TBIL	19.8 ± 10.0	21.3 ± 11.6	0.206
DBIL	7.19 ± 5.21	7.16 ± 4.43	0.948
Alb	36.9 ± 4.77	37.3 ± 4.75	0.539
Glob	28.8 ± 5.52	27.6 ± 5.78	0.067
GLR	67.3 ± 93.0	76.5 ± 129	0.478
ALP	89.8 ± 39.0	85.3 ± 31.0	0.221
Palb	136 ± 54.8	144 ± 63.8	0.207
PT	12.7 ± 1.55	12.8 ± 1.80	0.597
PTA	85.4 ± 15.4	85.5 ± 17.3	0.974
INR	1.12 ± 0.139	1.13 ± 0.160	0.733
APTT	33.7 ± 4.79	33.0 ± 5.00	0.168
Fib	2.81 ± 1.01	2.77 ± 0.79	0.109
TT	15.9 ± 2.25	16.0 ± 2.10	0.854
AFP	348 ± 2130	242 ± 1100	0.512

Abbreviation: BCLC: Barcelona Clinic Liver Cancer; T.N: tumor number; T.S: tumor size; WBC: leukocyte; NLR: neutrophil-to-lymphocyte ratio; MLR: monocyte-to-lymphocyte ratio; RBC: erythrocyte; Hb: hemoglobin; PLR: platelet-to-lymphocyte ratio; ALT: alanine aminotransferase; AST: aspartate aminotransferase; TBIL: total bilirubin; DBIL: direct bilirubin; Alb: albumin; Glob: globulin; GLR: globulin-to-lymphocyte ratio; ALP: alkaline phosphatase; Palb: prealbumin; PT: prothrombin time; PTA: prothrombin activity; INR: international normalized ratio; APTT: activated partial thromboplastin time; Fib: fibrinogens; TT: thrombin time; AFP: alpha-fetoprotein.

**Table 3 microorganisms-12-00976-t003:** Cox proportional hazard regression was employed for recurrence prediction based on LASSO regression.

	*p*-Value	HR	95%CI for HR	
			Lower	Upper
Age	0.177	1.012	0.995	1.029
Gender	0	0.451	0.295	0.69
BCLC	0	1.633	1.298	2.054
Ablative modality	0.101	0.759	0.545	1.056
GLR	0.307	1.001	0.999	1.002
Glob	0.015	1.032	1.006	1.059
MLR	0.02	2.27	1.14	4.522

Abbreviation: BCLC: Barcelona Clinic Liver Cancer; GLR: globulin-to-lymphocyte ratio; Glob: globulin; MLR: monocyte-to-lymphocyte ratio.

## Data Availability

The data that support the findings of this study are available from the corresponding author upon reasonable request.

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
