# Peer review of "The Effect of Low HBV-DNA Viral Load on Recurrence in Hepatocellular Carcinoma Patients Who Underwent Primary Locoregional Treatment and the Development of a Nomogram Prediction Model"

_microorganisms, 2024, doi:10.3390/microorganisms12050976_

Round 1
Reviewer 1 Report
Comments and Suggestions for Authors
The research article by Yiqi Xiong et al., entitled “Development and Validation of a Machine Learning-Based Nomogram to Predict the Recurrence of Hepatocellular Carcinoma Patients with Low HBV-DNA Viral Load in Primary Locoregional Treatment” aimed to investigate the prognostic implication of a low load of HBV DNA in HCC patients who underwent local treatment and also developed and validated a nomogram to predict the recurrence of patients with low (20-100 IU/mL) viral road (L-VL). The research article has interesting finding and may be useful for the management of HBV related HCC patients.
Minor Comments
1) Title of the table 1 <20 (N=72) <100 (N=403) P value, <100 (N=403) should be 20-100 (N=403) .
Author Response
Dear reviewer:
Thank you very much for your precious time and professional advice!
Minor Comments: Title of the table 1 <20 (N=72) <100 (N=403) P value, <100 (N=403) should be 20-100 (N=403) .
Response: Thank you very much for your reminder! We have modified the title of table1 at page5, line 197.
Reviewer 2 Report
Comments and Suggestions for Authors
What is your experience with screened HBV positive patients without HCC? Is there any evidence that those with low HBV levels are more predisposed to subsequently develop HCC in the future as was the case in your study?
Author Response
Dear reviewer:
Thank you very much for your precious time and professional advice!
Comments: What is your experience with screened HBV positive patients without HCC? Is there any evidence that those with low HBV levels are more predisposed to subsequently develop HCC in the future as was the case in your study?
Response: Several risk models have been constructed for the prediction of HCC development in HBV-related patients, including the D2AS model, CU-HCC model, AGED model, et al. In the D2AS model with HBV-DNA load, the results suggest that the higher the DNA levels, the greater the risk of patients developing HCC. Your advice also inspires us a lot. In the future, we will further explore the role of HBV-DNA load in the development of HCC. Thanks again for your valuable time and professional questions!